# The Link between Three Single Nucleotide Variants of the *GIPR* Gene and Metabolic Health

**DOI:** 10.3390/genes13091534

**Published:** 2022-08-26

**Authors:** Joanna Michałowska, Ewa Miller-Kasprzak, Agnieszka Seraszek-Jaros, Adrianna Mostowska, Paweł Bogdański

**Affiliations:** 1Department of Treatment of Obesity, Metabolic Disorders and Clinical Dietetics, Poznan University of Medical Sciences, 61-701 Poznan, Poland; 2Department of Bioinformatics and Computational Biology, Poznan University of Medical Sciences, 61-701 Poznan, Poland; 3Department of Biochemistry and Molecular Biology, Poznan University of Medical Sciences, 61-701 Poznan, Poland

**Keywords:** obesity, metabolic syndrome, *GIPR* gene, single nucleotide variant, GIP, metabolic health

## Abstract

Single nucleotide variants (SNVs) of the *GIPR* gene have been associated with BMI and type 2 diabetes (T2D), suggesting the role of the variation in this gene in metabolic health. To increase our understanding of this relationship, we investigated the association of three *GIPR* SNVs, rs11672660, rs2334255 and rs10423928, with anthropometric measurements, selected metabolic parameters, and the risk of excessive body mass and metabolic syndrome (MS) in the Polish population. Normal-weight subjects (*n* = 340, control group) and subjects with excessive body mass (*n* = 600, study group) participated in this study. For all participants, anthropometric measurements and metabolic parameters were collected, and genotyping was performed using the high-resolution melting curve analysis. We did not find a significant association between rs11672660, rs2334255 and rs10423928 variants with the risk of being overweight. Differences in metabolic and anthropometric parameters were found for investigated subgroups. An association between rs11672660 and rs10423928 with MS was identified. Heterozygous CT genotype of rs11672660 and AT genotype of rs10423928 were significantly more frequent in the group with MS (OR = 1.38, 95%CI: 1.03–1.85; *p* = 0.0304 and OR = 1.4, 95%CI: 1.05–1.87; *p* = 0.0222, respectively). Moreover, TT genotype of rs10423928 was less frequent in the MS group (OR = 0.72, 95%CI: 0.54–0.95; *p* = 0.0221).

## 1. Introduction

Obesity is a chronic metabolic disease caused by increased body fat stores [1,2]. This condition is associated with over 200 comorbidities, which include type 2 diabetes (T2D), cardiovascular disease (CVD), hypertension, obstructive sleep apnea, non-alcoholic fatty liver disease (NAFLD), and several types of cancer. Their occurrence may be a consequence of metabolic effects of excessive adipose tissue or increased body mass itself [3,4]. Obesity can be defined as Body Mass Index (BMI) ≥ 30 kg/m^2^, whereas a BMI of 25 to 29.9 kg/m^2^ indicates overweight [1]. Approximately 650 million adults worldwide were obese in 2016, and it is estimated that by 2030, 20% of the world’s population will be obese, and 38% will be overweight [4]. According to the Eurostat data, already 53% of the European population was overweight in 2019 [5]. The prevalence of excessive body mass is even higher in the United States, as data from National Health and Nutrition Examination Survey (NHANES) showed that 73.6% of adults were overweight in 2017–2018 [6]. The fundamental cause of obesity is an energy imbalance, i.e., a disproportion between calories consumed and expended. However, many other factors, besides diet and physical activity, can influence body weight. Those include other lifestyle components, such as sleep or stress level, social and economic determinants, developmental and gastrointestinal aspects, and genetic factors [2,4]. Several mutations in a single gene have been identified to be an underlying cause of obesity, e.g., a homozygous mutation in leptin or leptin receptor. Their prevalence is extremely rare and is characterized by an early onset of the disease [7]. Most often, obesity results from complex interactions among multiple genes and environmental factors that remain poorly understood. So far, over 1000 genes have been identified for polygenic obesity [8]. They were determined by genome-wide association studies (GWAS), which identify genomic variants statistically associated with the risk of a particular trait or disease [9].

*GIPR* (Gastric Inhibitory Polypeptide Receptor, OMIM * 137241) is one of the genes, which have been identified to be related to obesity risk in various populations [10,11,12,13,14,15]. It is located on chromosome 19 and encodes a receptor for glucose-dependent insulinotropic polypeptide (GIP). GIP is a 42-amino acid polypeptide, which is one of two known incretins—hormones secreted by the gastrointestinal tract [16]. The main role of this protein is the stimulation of insulin secretion from pancreatic β cells after meal consumption [17]. However, GIP receptors are expressed in various organs, including adipose tissue and the nervous and musculoskeletal system. Research showed that GIP, besides insulinotropic activity, may exert other functions, including promotion of fat storage in subcutaneous adipose tissue, supporting bone formation and limiting bone resorption, and regulation of energy balance [17,18].

Research shows that rs11672660, rs2334255, and rs10423928 single nucleotide variants (SNVs) are among *GIPR* variants associated with BMI, anthropometric measurements and metabolic health [19,20,21,22,23]. Rs11672660 (C > T) is an intronic variant with the minor allele frequency (MAF) of 0.214 (1000 Genomes; European population) [24,25]. It has been previously associated with BMI [20], however, Guo et al. confirmed only the nominal significance of this association [26]. So far, the relationship between this SNV and other anthropometric traits, metabolic parameters and metabolic syndrome (MS) has not been extensively studied, as there are only single studies available linking rs11672660 with lower high-density lipoprotein (HDL) and higher post-challenge glucose concentration [27]. Rs2334255 (G > T) is located in the 3′ untranslated region of the *GIPR* sequence [25]. The MAF of this variant for the European population is equal to 0.245 according to the 1000 Genomes project [24]. This SNV was identified as a common variant associated with both, obesity and T2D risk, in the analysis of two large datasets of GWAS [21]; however, the data describing the association of this variant with BMI, MS, anthropometric measurements and metabolic parameters are scarce. Rs10423928 (T > A) is an intronic variant with MAF equal to 0.214 (1000 Genomes; European population) [24,25]. Contrary to the two above-mentioned SNVs, this variant was associated with lower BMI and waist circumference (WC) [22,28]. Moreover, research linked rs10423928 with glycaemic traits and T2D [23,29].

To increase our understanding of the contribution of *GIPR* variants to metabolic health, the aim of our study was to (1) establish the association of three SNVs, rs11672660, rs2334255 and rs10423928, with the risk of excessive body mass and MS; and (2) investigate their association with anthropometric measurements and selected metabolic parameters in the Polish population.

## 2. Materials and Methods

### 2.1. Participants

A total of 940 adult subjects (≥18 years old) were enrolled in this study. Based on the BMI, they were assigned to one of the two groups. Control group (*n* = 340) included participants with BMI < 25 kg/m^2^, whereas subjects with excessive body mass (BMI ≥ 25 kg/m^2^) were allocated to the study group (*n* = 600). All participants received oral and written information about the research and signed an informed consent. All procedures performed in this study were approved by the Ethics Committee at Poznan University of Medical Sciences (approval no. 643/20).

### 2.2. Procedures

Anthropometric measurements and blood samples were collected from all patients. Based on the results, the presence of MS was determined using criteria described in the joint interim statement of the International Diabetes Federation (IDF); National Heart, Lung, and Blood Institute (NHLBI); American Heart Association (AHA); World Heart Federation (WHF); International Atherosclerosis Society (IAS); and International Association for the Study of Obesity (IASO) [30]. According to this document, MS can be identified when at least three out of the following five medical conditions are present: WC ≥ 80 cm for females and ≥94 cm for males; triglycerides (TG) ≥ 150 mg/dL (or treatment of hypertriglyceridemia); HDL ≤ 50 mg/dL for females and ≤40 mg/dL for males (or treatment for low HDL); systolic blood pressure (BP) ≥ 130 and/or diastolic BP ≥ 85 mm Hg (or treatment of hypertension); and fasting glucose ≥ 100 mg/dL (or pharmacological treatment of elevated glucose concentration). Due to the lack of required data, it was not possible to determine the MS status for 89 out of 940 participants.

#### 2.2.1. Anthropometric Measurements

Anthropometric measurements included body mass, waist and neck circumference, and height. To reduce the risk of measurement error, all participants underwent measurement procedures without shoes and while wearing light clothing. Body weight was measured to the nearest 0.1 kg, and height was measured to the nearest 1 cm using electronic scales with a stadiometer (Charder MS4900). Based on those two readings, BMI was calculated by dividing the body mass [kg] by the square of the body height [m]. Normal weight was defined as BMI within a range from 18.5 kg/m^2^ to 24.9 kg/m^2^, and excessive body weight was recognized when BMI was equal to or higher than 25 kg/m^2^ [31]. WC was measured at the middle point between the iliac crest and the lowest rib. Neck circumference (NC) was measured at a point just below the larynx and perpendicularly to the long axis of the neck. A certified tape measure (Seca 201) was used to measure both circumferences.

#### 2.2.2. Biochemical Assays

All subjects had venous blood collected from a cubital vein, while in the fasting state. The blood samples were centrifuged and frozen at −80 °C. The concentration of glucose, TG, HDL, alanine transaminase (ALT) and aspartate transaminase (AST) were determined using standardized commercial tests.

### 2.3. Genotyping of GIPR Single Nucleotide Variants

The salting out technique was used to isolate genomic DNA. High-resolution melting curve (HRM) analysis was performed to genotype three SNVs of the *GIPR* gene: rs11672660, rs2334255 and rs10423928. Light Cycler 96 system (Roche Diagnostics, Mannheim, Germany) and 5× HOT FIREPol EvaGreen HRM Mix (Solis BioDyne, Tartu, Estonia) were used for this analysis. HRM reaction conditions and primer sequences are available in the Appendix A. Samples that failed genotyping twice were excluded from the further analysis. For quality control, approximately 10% of randomly selected samples were regenotyped using the same genotype method.

### 2.4. Statistical Analysis

Statistical analysis was performed using Dell Statistica version 13 (2017, Tulsa, OK, USA). Continuous variables are summarized by median and mean ± standard deviation (SD). To examine if they were normally distributed in the population, Shapiro–Wilk normality test was initially performed. If the test did not confirm the normal distribution, non-parametric methods were used for the statistical analysis. The results for the control and the study group were compared using the Mann–Whitney test to verify differences. The Kruskal–Wallis test was used to compare the differences among particular genotypes of GIPR variants. If statistically significant differences were found, a post-hoc Dunn’s test was implemented. Categorial variables are presented using frequency and percentages. Odds ratios (ORs) and 95% confidence intervals (CIs) were calculated. To investigate if any of the studied SNVs is associated with a higher or lower risk of excessive body weight and/or MS, the logistic regression was performed using MedCalc^®^ Statistical Software version 20.027 (MedCalc Software Ltd., Ostend, Belgium). For all described tests, results were considered significant for *p*-value < 0.05. Hardy-Weinberg (HW) equilibrium was evaluated for every SNV using Chi-square (χ^2^) test. All assessed variants were in HW equilibrium (*p*-value > 0.05).

## 3. Results

The data of 940 subjects were included in the analysis. Control group with BMI < 25 kg/m^2^ consisted of 340 participants and the remaining 600 patients were included in the study group with excessive body mass (BMI ≥ 25 kg/m^2^). In the study group, 237 participants were obese (≥30 kg/m^2^). Both groups presented statistically significant differences for all continuous variables measured. The mean BMI in the control group was equal to 22.3 kg/m^2^ (±SD 1.79) and 29.7 kg/m^2^ (±SD 4.06) in the study group, whereas the mean weight was 62.2 kg (±SD 8.7) and 82.3 kg (±SD 13.3), respectively. The mean age was lower in the control group than in the study group (48 years old ± SD 14.5 vs. 57 years old ± SD 13.2). There were more men (38.5%, *n* = 231) in the study group than in the control group (21%, *n* = 71). Criteria for MS were met by 52% (*n* = 288) of participants from the study group and 13% (*n* = 38) of the control group. The detailed characterization of the control and the study group is presented in Table 1.

For each SNV, there was no statistically significant deviation from the Hardy–Weinberg equilibrium for both, the control and the study group. The following number of samples failed the genotyping: rs11672660—32 samples (0.03%); rs2334255—26 samples (0.03%); and rs10423928—12 samples (0.01%). Statistical analysis of the distribution of the particular alleles and genotypes of rs11672660, rs2334255 and rs10423928 did not show any differences between the control and the study group. The data is presented in Table 2.

Statistical analysis of the distribution of the particular alleles and genotypes of rs2334255 in subjects with or without MS also did not show statistically significant differences between groups. However, the differences were detected for genotypes of rs11672660 and rs10423928 variants. For all SNVs, there was no statistically significant deviation from the Hardy–Weinberg equilibrium for either group. There was a significant difference in the genotype distribution of rs11672660, where heterozygous CT genotype was significantly more frequent in the group with MS (OR = 1.38, 95%CI: 1.03–1.85; *p* = 0.0304). Heterozygous AT genotype of rs10423928 was also more frequent in the group with MS (OR = 1.4, 95%CI: 1.05–1.87; *p* = 0.0222). On the contrary, homozygous TT genotype of this variant was less frequent in this group (OR = 0.72, 95%CI: 0.54–0.95; *p* = 0.0221). The genotype distribution of rs11672660, rs2334255 and rs10423928 in the subjects with and without MS is presented in Table 3.

Results of performed logistic regression showed that rs11672660, rs2334255 and rs10423928 are not associated with elevated or lower risk of excessive body mass and MS. The results did not change after adjusting the model for sex and age. They are presented in Table 4.

To investigate if any particular genotype of rs11672660, rs2334255 and rs10423928 is associated with anthropometric measurements and selected metabolic parameters, we compared the age, body mass, BMI, WC, NC, and glucose, ALT, AST, TG and HDL concentration in every genotype group. The analysis did not show any differences between TT, CT and CC carriers of rs11672660 for those parameters, nor for TT, GT and GG carriers of rs2334255. For the rs10423928 variant, the differences between AA, AT and TT carriers were also not found. The results are available in Appendix A. Furthermore, anthropometric measurements and metabolic parameters were compared for all genotypes within four subgroups: control and the study group, and the group with and without MS. The following parameters were compared: age, body mass, BMI, WC, NC, and glucose, TG, HDL, AST and ALT concentration. There were no differences between TT, CT and CC carriers of rs11672660 in the study group, control group and the group without MS. In the group with MS, CT carriers had lower AST (27.09 IU/L ± SD = 8.71; *p* = 0.0033) and ALT (32.02 IU/L ± SD = 16.72; *p* = 0.0457) concentrations than CC carriers (31.07 IU/L ± SD = 13.34 and 36.99 IU/L ± SD = 20.96, respectively). All results of performed analyses are available in Appendix A. For rs2334255 there were no differences between TT, GT and GG carriers in the control group and the group with MS. In the study group, TT carriers had lower ALT concentration than GT carriers (27.43 IU/L ± SD = 12.39 vs. 37.27 IU/L ± SD = 26.04, respectively; *p* = 0.0419). In the group without MS, GT carriers had lower body mass (68.99 kg ± SD = 13.41; *p* = 0.0461) and NC (34.89 cm ± SD = 3.47 IU/L; *p* = 0.0398) than GG carriers (72.31 kg ± SD = 14.83 and 35.86 kg ± SD = 3.87, respectively). Described results are available in Appendix A. There were no differences between AA, AT and TT carriers of rs10423928 in the study and the control group, and in the group without MS. In the group with MS, AT carriers had lower AST (27.1 IU/L ± SD = 8.64 IU/L; *p* = 0.0022) and ALT (31.77 IU/L ± SD = 16.59; *p* = 0.0113) concentrations than TT carriers (31.39 IU/L ± SD = 13.93 and 37.17 IU/L ± SD = 20.55, respectively). All results of performed analyses are available in Appendix A.

## 4. Discussion

In our study, we did not find any differences in the distribution of the particular alleles and genotypes of rs11672660, rs2334255 and rs10423928 variants between the control group and the study group with the excessive body mass. The logistic regression analysis also did not find any genotypes associated with a higher or lower risk of excessive weight. This is contrary to the available research, as rs11672660, rs2334255, and rs10423928 have been previously associated with BMI. Analysis of the data from the Genetic Investigation of Anthropometric Traits (GIANT) Consortium identified rs11672660 as a variant associated with BMI [20]. Moreover, conditional quantile regression performed on the sample BMI distribution in 75,230 adults of European ancestry recognized rs11672660 as one of the nine SNVs, which effect increased significantly across the sample BMI distribution. Authors concluded that gene-gene and gene-environment interactions play an important role in shaping the genetic architecture of BMI [32]. On the contrary, gene-centric meta-analyses of 108 912 individuals of European ancestry showed nominal significance for an association of this variant with BMI [26], whereas Scott et al. found a negative correlation between rs11672660 variant occurrence and BMI [27]. The inconsistencies between the studies might be associated with complex gene-environment and gene-gene interactions influencing BMI. Various obesity-predisposing gene variants have been described to interact with lifestyle factors, such as diet, physical activity, sleep duration, or alcohol consumption [33]. Research showed that the impact of variant rs9939609 in *FTO* (FTO α-ketoglutarate dependent dioxygenase, OMIM * 610966) on BMI may be minimal in lean populations, where excessive food is scarce, compared to populations where food is easily accessible [34]. Moreover, a higher intake of fried foods has been shown to increase the impact of the 32 SNV gene score on BMI [35]. Therefore, SNVs could potentially affect BMI through a mixture of genetic and environmental interactions, which may differ for populations. This could also explain the lack of association between studied variants and excessive weight as our study included only Polish subjects.

In the study of Zhang et al. where the genetic-pleiotropy-informed conditional false discovery rate approach was used, rs2334255 was identified as a novel common variant associated with both, obesity and T2D risk [21]. Another study investigating the relationship of this variant with adiposity traits was an analysis of Wang et al. Only samples from the Han Chinese population had been included, and the results showed that rs2334255 was neither associated with the visceral fat area nor with subcutaneous area [36]. The association of this SNV with other anthropometric measurements and metabolic parameters is not available. Our results showed that in the subject without MS, GT carriers of this variant have lower body mass and neck circumference, suggesting that particular genotypes of this variant might be associated with anthropometric measurements, but only in specific subgroups. Moreover, in the study group with excessive body mass, a different association was found, where GT carriers had higher ALT concentrations than TT carriers. Excessive body mass had been reported to be a major risk factor for elevated ALT activity, and the hepatoxic effect of visceral adipose deposition has been proposed as one of the potential mechanisms of this association [37]. Additionally, liver enzyme levels may be mediated by the biological effects, which are related to genetic and environmental factors, including alcohol consumption, smoking, and coffee consumption [38]. Our study indicates that genetic variation may be associated with the additional risk of elevated liver enzymes, however, this association requires further investigation.

Rs11672660 variant has been previously associated with glucose metabolism. The Meta-Analyses of Glucose and Insulin-related traits Consortium (MAGIC) found the association of this variant with post-challenge glucose concentration (2-h glucose) [23]. The results were replicated by Scott et al. Moreover, their research linked the rs11672660 variant with an increased risk of T2D and lower HDL concentration [27]. In our study, only data on fasting glucose was available, therefore it was not possible to investigate the association of the rs11672660 variant with post-challenge glucose concentration. Moreover, our results showed the lack of association of this variant with TG and HDL. When the concentrations of aminotransferases were investigated, the only statistically significant result found was for CT carriers, who met the criteria for MS, as they were characterized by the lower AST and ALT in comparison to homozygous CC carriers. Concurrently, the CT genotype of the rs11672660 variant was associated with a higher risk of MS in our study, and this particular condition has been reported to be a risk factor for elevated aminotransferases level [39]. The possible explanation for the opposite association found in this study is variation in ALT and AST activity caused by many factors. Extreme physical exertion, viral hepatitis, alcohol consumption, medication, and demographic factors can interfere with aminotransferases levels [37,40]. To our knowledge, this is the only study investigating the relationship between the rs11672660 variant and described metabolic parameters. Further research is needed to establish the association between this SNV and glucose, TG, HDL, AST and ALT concentration. Future studies will help to establish potential mechanisms in which different genetic variants of rs11672660 may positively or negatively impact metabolic health.

In our analysis, we did not find the association of rs10423928 with body mass and anthropometric measurements. Similar results were obtained by Zhang et al. as in the study of postmenopausal women in Shanghai, the researchers also did not find a relationship between rs10423928 with BMI [41]. Moreover, Barbosa-Yanez et al. did not observe differences in BMI, the content of body fat and liver fat measured by magnetic resonance imaging and spectroscopy between risk allele A carriers and homozygous carriers of major allele T of this variant [42]. This is contrary to the results of Lyssenko et al., as their analysis of data from 11 studies showed that the A allele of this variant was associated with a decrease in BMI and WC [22]. Moreover, a meta-analysis of 22 studies including 54,884 nondiabetic individuals revealed that rs10423928 was associated with a 0.11 kg/m^2^ lower BMI per allele [28]. Ahlqvist et al. showed that the A allele is associated with lower BMI, better insulin sensitivity and lower adipose tissue osteopontin mRNA levels. Osteopontin plays a crucial role in adipose tissue subclinical inflammation and its’ higher levels were associated with insulin resistance characteristic of obesity. Authors suggested that carriers of the minor allele of rs10423928 are characterized by the reduced GIP receptor function [43]. Despite the fact that this variant is located within a noncoding region, an intronic gene variant can still shift gene expression by affecting gene splicing, transcription and translation [44]. *GIPR* encodes the GIP receptor, which has been previously associated with adiposity in animals and humans [45]. The research showed that *GIPR* knockout mice are prevented from developing obesity induced by a high-fat diet (HFD) [46].

We did not find the association of rs10423928 with fasting glucose, TG and HDL concentration. Stancakova et al. also did not find the association of this variant with HDL concentration in their study on the effect of 34 risk loci associated with T2D and hyperglycaemia on lipoprotein subclasses in nondiabetic Finnish men [47]. However, this variant was associated with glycaemic traits in several studies. Barbosa-Yanez et al. showed that in the group of diabetic and prediabetic subjects, A allele carriers of this variant had increased fasting glucose and lower glucose levels 2 h after an oral glucose challenge [42]. This result was opposite to large meta-analyses, which showed that this common variant is associated with higher glucose and lower insulin levels after the oral glucose challenge test and with a diminished incretin effect [23,29]. An analysis of 11 studies showed that the A allele of this variant is associated with impaired glucose and GIP-stimulated insulin secretion. As described above, in this study, A carriers were also characterized by lower BMI, which neutralized the effect of impaired insulin secretion on the risk of T2D [22]. GIP is one of the incretins, which stimulates insulin secretion after oral glucose intake and is responsible for the regulation of glucose homeostasis. Therefore, the receptor of this hormone—GIPR might be a candidate for mediating insulin secretion after the oral glucose challenge. It has been suggested that alterations in this receptor might be associated with T2D pathophysiology [16]. Moreover, Lyssenko et al. linked rs10423928 with decreased GIPR and osteopontin expression, which resulted in reduced β cell proliferation and increased apoptosis [22]. Additionally, gene-diet interactions have been reported to contribute to the development of T2D. The study by Sonestedt et al. reported significant interactions between rs10423928 and fat and carbohydrate intake in relation to T2D risk. According to their results, AA genotype carriers who consumed a high-fat and low-carbohydrate diet, had reduced T2D risk, whereas a diet high in carbohydrates and low in fat was more beneficial for TT genotype carriers [48]. On the contrary, an analysis of data from European Prospective Investigation into Cancer (EPIC)-InterAct did not detect a significant interaction between rs10423928 and carbohydrate or fat intake for T2D risk [49]. A large study of gene-lifestyle interactions confirmed the association of rs10423928 with 2-h glucose; however, no interactions between genetic and lifestyle factors were found, suggesting that studied variants do not exhibit strong subgroup-specific effects [28].

In our study, we showed the differences in the distribution of the genotypes of two *GIPR* variants, rs11672660 and rs10423928, in subjects with and without MS suggesting that those variants are associated with MS risk. However, this association was not confirmed by the logistic regression. To our knowledge, this is the first study to find the link between those variants and the MS criteria. In the studied Polish population, the homozygous TT allele of the rs10423928 variant was associated with a lower risk of MS. As described above, available research suggests that the A allele of this variant is associated with worse glycaemic traits and T2D risk. Glucose concentration (or treatment of impaired glucose) is one of the criteriums of MS, therefore impairments in glucose homeostasis could increase the risk of meeting MS criteria. In our study, we did not find an association of this variant with glucose concentration. This could be associated with the limitations of our study as only fasting measurement was available, and glucose challenge was not performed. Moreover, the fat distribution measurements were not collected. Due to the large sample size, single parameters and variables were not available for all subjects. Consequently, determination of MS status was not possible for some participants. Moreover, the genotype data were not available for the whole dataset, as part of the samples failed the genotyping, and they were excluded from further statistical analysis.

## 5. Conclusions

In conclusion, we did not find a significant association between rs11672660, rs2334255 and rs10423928 and the risk of excessive body mass. Several correlations between studied variants and metabolic parameters were identified in investigated subgroups. The novel association of two *GIPR* variants, rs11672660 and rs10423928, with MS was identified. Heterozygous CT genotype of rs11672660 and AT genotype of rs10423928 were significantly more frequent in the group with MS. Moreover, the homozygous TT genotype of the rs10423928 variant was less frequent in the MS group. Therefore, our results suggest that genetic variation in *GIPR* may be linked with MS. A better understanding of the risk factors of this complex condition is of great clinical interest, as their early identification could prevent the development of MS comorbidities, which include CVD, T2D and other health problems. Future studies are needed to investigate the relationship of studied SNVs with MS and metabolic parameters to explain the underlying mechanisms and elucidate the importance of genetic variation in *GIPR* in metabolic health.

## Figures and Tables

**Table 1 genes-13-01534-t001:** The detailed characterization of the control and the study group.

	Control Group				Study Group				*p*-Value
Parameters	Mean	Median		SD	Mean	Median		SD	
Age [years]	48.42	47.00		14.52	56.88	60.00		13.19	<0.0001
Body weight [kg]	62.15	60.60		8.72	82.29	81.50		13.32	<0.0001
BMI [kg/m^2^]	22.33	22.51		1.79	29.72	28.72		4.06	<0.0001
WC [cm]	79.76	79.00		9.28	99.42	99.00		11.97	<0.0001
NC [cm]	33.81	33.00		3.03	37.62	38.00		3.52	<0.0001
Glucose [mg/dL]	90.20	88.00		16.57	98.40	92.00		27.90	<0.0001
TG [mg/dL]	111.90	94.00		69.12	170.39	142.50		121.50	<0.0001
HDL [mg/dL]	73.15	70.00		17.96	60.55	59.00		15.36	<0.0001
AST [IU/L]	26.26	25.00		7.64	29.48	27.00		12.97	<0.0001
ALT [IU/L]	24.33	22.00		11.63	34.97	29.00		23.59	<0.0001
	**Health status and tobacco use**								
	N		%		N		%		
Metabolic syndrome	38		13		288		52		<0.0001
Diabetes	16		5		76		13		<0.0001
Hypertension	72		22		283		48		<0.0001
CVD	25		8		65		11		0.0836
Cigarette smoker	54		16		100		17		0.7594

SD: standard deviation; BMI: Body Mass Index; WC: waist circumference; NC: neck circumference; TG: triglycerides; HDL: high-density lipoprotein; AST: aspartate transaminase; ALT: alanine transaminase; CVD: cardiovascular disease.

**Table 2 genes-13-01534-t002:** The distribution of particular alleles and genotypes in the study and the control group.

	Study Group *n* (%)	Control Group *n* (%)	OR (95% CI)	*p*-Value
**rs11672660**				
*allele*				
*T*	272 (23)	156 (24)	0.99 (0.79–1.24)	0.9541
*C*	886 (77)	502 (76)	1.01 (0.81–1.27)	
*genotypes*				
*TT*	34 (6)	18 (5.5)	1.08 (0.60–1.94)	0.8824
*CT*	204 (35)	120 (36.5)	0.95 (0.71–1.26)	0.7190
*CC*	341 (59)	191 (58)	0.96 (0.73–1.27)	0.8315
*pHW*	0.6346	0.8807		
**rs2334255**				
*allele*				
*T*	247 (21)	148 (23)	0.92 (0.73–1.15)	0.4773
*G*	925 (79)	508 (77)	1.09 (0.87–1.37)	
*genotypes*				
*TT*	25 (4)	11 (3)	1.28 (0.62–2.65)	0.5961
*GT*	197 (34)	126 (39)	0.81 (0.61–1.08)	0.1497
*GG*	364 (62)	191 (58)	1.18 (0.89–1.55)	0.2591
*pHW*	0.7985	0.0719		
**rs10423928**				
*allele*				
*A*	282 (24)	167 (20)	1.24 (0.99–1.54)	0.0588
*T*	908 (76)	666 (80)	0.81 (0.65–1.00)	
*genotypes*				
*AA*	37 (6)	23 (7)	0.89 (0.52–1.53)	0.6786
*AT*	208 (35)	121 (36)	0.94 (0.71–1.25)	0.7206
*TT*	350 (59)	189 (57)	1.09 (0.83–1.43)	0.5791
*pHW*	0.4161	0.5475		

pHW: Hardy–Weinberg *p*-value.

**Table 3 genes-13-01534-t003:** The genotype distribution of rs11672660, rs2334255 and rs10423928 in the subjects with and without metabolic syndrome.

	With MS *n* (%)	Without MS *n* (%)	OR (95% CI)	*p*-Value
**rs11672660**				
*allele*				
*T*	162 (26)	230 (22)	1.20 (0.95–1.51)	0.1370
*C*	472 (74)	802 (78)	0.84 (0.66–1.05)	
*genotypes*				
*TT*	18 (6)	31 (6)	0.84 (0.52–1.71)	0.8807
*CT*	126 (40)	168 (33)	1.38 (1.03–1.85)	0.0304
*CC*	173 (54)	317 (61)	0.75 (0.57–1.00)	0.0593
*pHW*	0.4258	0.1723		
**rs2334255**				
*allele*				
*T*	135 (21)	231 (22)	0.94 (0.74–1.19)	0.6266
*G*	505 (79)	811 (78)	1.06 (0.84–1.35)	
*genotypes*				
*TT*	10 (3)	27 (5)	0.59 (0.28–1.24)	0.1704
*GT*	115 (36)	177 (34)	1.09 (0.81–1.46)	0.6016
*GG*	195 (61)	317 (61)	1.04 (0.75–1.34)	1.0000
*pHW*	0.1546	0.7232		
**rs10423928**				
*allele*				
*A*	171 (26)	237 (22)	1.25 (0.99–1.57)	0.0536
*T*	475 (74)	825 (78)	0.79 (0.64–1.00)	
*genotypes*				
*AA*	21 (6.5)	33 (6)	1.05 (0.60–1.85)	0.8853
*AT*	129 (40)	171 (32)	1.40 (1.05–1.87)	0.0222
*TT*	173 (53.5)	327 (62)	0.72 (0.54–0.95)	0.0221
*pHW*	0.6408	0.1008		

MS: Metabolic Syndrome; pHW: Hardy–Weinberg *p*-value.

**Table 4 genes-13-01534-t004:** Logistic regression investigating the association of rs11672660, rs2334255 and rs10423928 variants with excessive weight and metabolic syndrome.

		Excessive Weight ^1^				Metabolic Syndrome			
		OR	95% CI	OR	95% CI	OR	95% CI	OR	95% CI
		*CT* ^2^		*CC* ^2^		*CT* ^2^		*CC* ^2^	
**rs11672660**	crude	0.9	0.54–1.48	0.94	0.58–1.53	1.37	0.82–2.28	0.99	0.61–1.63
	adjusted ^3^	1.15	0.67–1.95	1.26	0.75–2.11	1.7	0.99–2.92	1.17	0.70–1.97
		*GT* ^2^		*GG* ^2^		*GT* ^2^		*GG* ^2^	
**rs2334255**	crude	0.92	0.53–1.62	1.12	0.65–1.94	1.59	0.87–2.88	1.5	0.84–2.67
	adjusted ^3^	0.92	0.51–1.69	1.1	0.62–1.97	1.71	0.91–3.19	1.56	0.85–2.87
		*AT* ^4^		*TT* ^4^		*AT* ^4^		*TT* ^4^	
**rs10423928**	crude	1.23	0.73–2.06	1.32	0.80–2.18	1.16	0.67–1.99	0.81	0.48–1.38
	adjusted ^3^	1.27	0.74–2.21	1.43	0.84–2.43	1.23	0.69–2.18	0.83	0.47–1.43

OR: odds ratio; CI: confidence interval; ^1^ BMI ≥ 25 kg/m^2^; ^2^ TT carriers used as a reference, ^3^ adjusted for sex and age; ^4^ AA carriers used as a reference.

## Data Availability

The genotype dataset presented in this study can be found in European Variation Archive online repository under the following accession number: PRJEB54891.

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
