# Peer review of "The Link between Three Single Nucleotide Variants of the GIPR Gene and Metabolic Health"

_genes, 2022, doi:10.3390/genes13091534_

Round 1
Reviewer 1 Report
This is an original research article, which aims to investigate the association of three SNVs, rs11672660, rs2334255 and rs10423928, with the risk of excessive body mass and MS; as well as their association with anthropometric measurements and selected metabolic parameters in the Polish population.
Generally, the topic is quite interesting, and the authors have in depth knowledge. They have used the appropriate methodology, study design, and adequate statistical analysis. The results are sufficiently well-presented, clear, and easy to understand, so as to reach safe and solid conclusions. Overall, the manuscript is well written and structured. Thus, I think it would make a nice addition to Genes as an original research article.
However, the following points should be generally considered, thus minor revision is demanded.
1. Line 79-81: Please refer the existing data according to current literature.
2. Line 114: Were data as waist-hip ratio, body fat distribution and other fat-related measurements collected?
3. Line 169: Please add more baseline characteristics, such as presence of type 2 diabetes, tobaco use, dyslipidemia, hypertension, CVD etc, if available.
4. Please refer the percentage of patients included in the study group with a BMI above 30 km/m2.
5. Line 204: “Logistic regression investigating the association of rs11672660, rs2334255 and rs10423928 variants with excessive weight (overweight) and metabolic disorder.” What about obese participants?
6. Line 221-223: Please further discuss this result.
7. Line 240-241: Please provide a possible explanation.
8. An extra paragraph discussing future perspectives and clinical implications would be a nice addition to the manuscript.
Author Response
Dear Reviewer,
Thank you very much for your relevant comments. All of them have been addressed to improve our manuscript. Please, find the summary below:
- The references for 1000 genes project and The Sequence Ontology have been added as the source of the MAF and consequence details of studied SNVs.
- Unfortunately, hip circumference (hence, waist-hip ratio) and body fat distribution were not collected. We are aware that this data would be of great interest for this study, and as it is not available, the relevant sentence was added to the study limitations section.
- More information regarding health status of the participants was added to the Table 1.
- The information about the number of participants with obesity was added in the lines 161-162.
- Apologies for the confusion, the wrong term was used in the Table 4. Logistic regression was investigating the association of studied SNVs with excessive weight, i.e. BMI≥25. Alterations in the table were made to reflect the text.
- The discussion regarding association of rs11672660 with aminotransferases levels was extended.
- The possible explanation was added at the end of the first paragraph of the discussion.
- An extra paragraph was added in the conclusion.
Reviewer 2 Report
The manuscript “ The link between three single nucleotide variants of the GIPR 2 gene and metabolic health” is interesting and tidy written, I have only a few details to ask:
here below my comments:
<<The Kruskal–Wallis test was used to compare the differences between particular genotypes of GIPR variants>>, The procedure is used to compare three or more groups on a dependent variable; should be AMONG instead of between.
I suggest typing clearly, better in table 1, how many samples failed the genotyping.
<<In conclusion, we did not find a significant association between rs11672660, 365 rs2334255 and rs10423928 and the risk of excessive body mass>>
could this be specific to the examined population?
The reference for the 1000 genome project is missing:
The 1000 Genomes Project Consortium. A global reference for human genetic variation. Nature 526, 68–74 (2015). https://doi.org/10.1038/nature15393
And I would suggest you read Lagou, V., Mägi, R., Hottenga, J.J. et al. Sex-dimorphic genetic effects and novel loci for fasting glucose and insulin variability. Nat Commun 12, 24 (2021). https://doi.org/10.1038/s41467-020-19366-9; to expand the discussion on sex-adjusted results.
Author Response
Dear Reviewer,
Thank you very much for your relevant comments. All of them have been addressed to improve our manuscript. Please, find the summary of the changes below:
- The word “between” was replaced by “among” in the sentence related to Kruskal-Wallis test.
- The number of samples that failed genotyping were incorporated into Table 2.
- The possible explanation of the lack of significant association between rs11672660, rs2334255 and rs10423928 and the risk of excessive body mass was added at the end of the first paragraph of the discussion.
- The reference for the 1000 genome project was added where necessary.
- Thank you very much for recommending very interesting work of Lagou et al. We deeply regret that the insulin and HOMA-IR were not determined for our study population. However, we are planning additional analyses of our data in the future to explore sex dimorphism in the traits available for our dataset.